Evidence for a non-linear carbon accumulation pattern along an Alpine glacier retreat chronosequence in Northern Italy

http://orcid.org/0000-0003-2957-9071 Montagnani Leonardo 1 2 leonardo.montagnani@unibz.it
Badraghi Aysan 1
Speak Andrew Francis 1
http://orcid.org/0000-0001-6994-274X Wellstein Camilla 1
Borruso Luigimaria 1
Zerbe Stefan 1
Zanotelli Damiano 1
1 Faculty of Science and Technology, Free University of Bozen-Bolzano , Bolzano , Italy
2 Forest Services, Autonomous Province of Bolzano , Bolzano , Italy
Cappelletti David
Electronic publication date: 2019 Oct 10
Publication date: 2019
Volume: 7
Electronic Location ID: e7703
Received 2019 Apr 30; Accepted 2019 Aug 19
Copyright: © 2019 Montagnani et al.
Copyright year: 2019
Copyright holder: Montagnani et al.
License: This is an open access article distributed under the terms of the Creative Commons Attribution License, which permits unrestricted use, distribution, reproduction and adaptation in any medium and for any purpose provided that it is properly attributed. For attribution, the original author(s), title, publication source (PeerJ) and either DOI or URL of the article must be cited.
License URL: https://creativecommons.org/licenses/by/4.0/

Keywords: Glacier retreat, Carbon accumulation, Plant colonization, Gas exchange, Soil analysis, Ecosystem respiration, Net ecosystem exchange, Gross ecosystem exchange, Life form

Funding: Stifterverband and the Foundation of the Free University of Bozen‐Bolzano, and by the UNIBZ internal project entitled “The influence of vegetation on carbon fluxes and soil carbon accumulation after glacier retreat,” CUP I41J13000070005 Open Access Publishing Fund of the Free University of Bozen-Bolzano This research was funded by the Dr. Erich‐Ritter‐Stiftung and Dr. Herzog‐Sellenberg‐ Stiftung within Stifterverband and the Foundation of the Free University of Bozen‐Bolzano, and by the UNIBZ internal project entitled “The influence of vegetation on carbon fluxes and soil carbon accumulation after glacier retreat,” CUP I41J13000070005. This work was supported by the Open Access Publishing Fund of the Free University of Bozen-Bolzano. The funders had no role in study design, data collection and analysis, decision to publish or preparation of the manuscript.

==============================
Background

The glaciers in the Alps, as in other high mountain ranges and boreal zones, are generally retreating and leaving a wide surface of bare ground free from ice cover. This early stage soil is then colonized by microbes and vegetation in a process of primary succession. It is rarely experimentally examined whether this colonization process is linear or not at the ecosystem scale. Thus, to improve our understanding of the variables involved in the carbon accumulation in the different stages of primary succession, we conducted this research in three transects on the Matsch glacier forefield (Alps, N Italy) at an altitude between 2,350 and 2,800 m a.s.l.

Methods

In three field campaigns (July, August and September 2014) a closed transparent chamber was used to quantify the net ecosystem exchange (NEE) between the natural vegetation and the atmosphere. On the five plots established in each of the three transects, shading nets were used to determine ecosystem response function to variable light conditions. Ecosystem respiration (Reco) and gross ecosystem exchange (GEE) was partitioned from NEE. Following the final flux measurements, biometric sampling was conducted to establish soil carbon (C) and nitrogen (N) content and the biomass components for each transect.

Results

A clear difference was found between the earlier and the later successional stage. The older successional stages in the lower altitudes acted as a stronger C sink, where NEE, GEE and Reco were significantly higher than in the earlier successional stage. Of the two lower transects, the sink capacity of intermediate-succession plots exceeded that of the plots of older formation, in spite of the more developed soil. Total biomass (above- and belowground) approached its maximum value in the intermediate ecosystem, whilst the later stage of succession predominated in the corresponding belowground organic mass (biomass, N and C).

Outlook

We found that the process of carbon accumulation along a glacier retreat chronosequence is not linear, and after a quite rapid increase in carbon accumulation capacity in the first 150 years, in average 9 g C m−2 year−1, it slows down, taking place mainly in the belowground biomass components. Concurrently, the photosynthetic capacity peaks in the intermediate stage of ecosystem development. If confirmed by further studies on a larger scale, this study would provide evidence for a predominant effect of plant physiology over soil physical characteristics in the green-up phase after glacier retreat, which has to be taken into account in the creation of scenarios related to climate change and future land use.

Introduction

Air temperature is increasing globally and glacier retreat is among the most striking evidence of global change (Walker & Del Moral, 2003; Nakatsubo et al., 2005; Barry, 2006; Kozioł et al., 2014; Pepin et al., 2015), and this is particularly evident in the Alps. Due to increases of annual mean temperature, 75% of all Alpine glaciers are estimated to be retreating in the last 150 years, exposing the abundant bare substrate to plant colonization (Walker & Del Moral, 2003). Glacial expansion in the Alps peaked around 1860 (Solomina et al., 2016), and since then glaciers have been retreating to higher altitudes, with an estimate rate of 150 m for 1 °C of temperature increase (Knoll & Kerschner, 2009). The retreat occurs also in other mountain ranges and boreal zones (Zemp et al., 2015; Sigl et al., 2018) leaving an abundance of bare ground free from ice cover. This emergent territory represents a new habitat for the biotic communities, i.e., bacteria, fungi, plants and animals. The starting colonization by these taxa and their subsequent development can have direct effects on ecosystem functioning and element cycling.

The formation of the new biocenosis above former bare soil is defined as a process of primary succession (Odum, 1969; Walker & Del Moral, 2011). In this process, microbes, vegetation and fauna are involved in the development of soil and in the accumulation of organic carbon (Matthews, 1992; Nakatsubo et al., 1998, 2005; Egli, Fitze & Mirabella, 2001; Bekku et al., 2004; Burga et al., 2009; Yoshitake et al., 2011; D’Amico et al., 2014; Li et al., 2018). The soil, with its properties tied to its developmental stage, acts as a major player in the carbon accumulation process, with its increasing potential for carbon release as the carbon stock increases but also with its increasing capacity for exchanging ions and sustaining the functions of photosynthetically active plants. Therefore, it represents a relevant variable when the formation of a new vegetation community is considered in a biogeochemical perspective.

The exposed terrain offers a mitigation potential for the global change, since the carbon sequestered by the newly established vegetation represents a negative feedback to the atmospheric increase of carbon dioxide and the consequent radiative forcing. Besides the global relevance of ongoing earth deglaciation, it is scarcely defined which are the main variables involved in the process of vegetation regrowth. Different biological factors, including vegetation and microbe type, and the environmental constraints, in particular temperature, radiation and snow cover (Desai et al., 2016; Metcalfe, Fisher & Wardle, 2011) should be taken into account to explain the sink capacity along the different phases of vegetation regrowth. All of these variables interact with the chemical and physical properties of the soil and local microclimatic conditions and with the effect of human activity and released carbon and nitrogen (Bradley, Anesio & Arndt, 2017).

An increasing number of studies have been undertaken in recently deglaciated terrain (Nakatsubo et al., 1998, 2005; Uchida et al., 2002, 2006; Bekku et al., 2004; Yoshitake et al., 2007; Guelland et al., 2012; Chae et al., 2016). However, a gap in our current understanding exists around the linkage between current vegetation gas exchange and achieved level of soil evolution and overall carbon accumulation. Plants in the different phases of primary succession frequently also show different lifeforms: While the earliest stages are dominated by mosses and annual herbs (Raunkiaer Therophytes) at the final phase forbs with abundant underground lignified tissue (Hemi- and Cryptophytes) prevail. However, do these life forms and their functional traits have any influence on the velocity with which the carbon in the soil is accumulated?

Among the several temporal patterns of carbon dynamics that the newly established ecosystem can show (Caccianiga et al., 2006), we can hypothesize three different idealized scenarios (Fig. 1). In scenario 0, both assimilation and respiration linearly increase along the chronosequence. In scenario 1, the physico–chemical characteristics of the soil and their positive feedback on assimilation capacity prevail, leading to a maximum sink capacity in the later stage of the chronosequence. In scenario 2, the effect of growth form and plant strategy prevails, leading to an assimilation and respiration pattern with a maximum sink at an intermediate stage.

Figure 1 A conceptual representation of the development of a primary succession.

(A) In Case 0, photosynthesis (A) and respiration (R) increase linearly. (B) In Case 1, photosynthesis and respiration increase less than linearly in the intermediate stage. (C) In Case 2, photosynthesis and respiration peak occur in the intermediate stage.

The scenario 0 is used in the largely deployed biogeochemical models, like Biome-BGC (Running & Gower, 1991; Thornton & Rosenbloom, 2005). In these models, a spin-up phase lasting hundreds of years is used to bring the ecosystem to stable conditions. In this scenario, the impact of vegetation type variation is not taken into account and the variation in carbon and nitrogen is considered to influence photosynthesis and respiration equally.

In scenario 1, we hypothesize that soil evolution is more relevant for the fluxes than vegetation characteristics. Plant C and soil organic C are in active exchange with the atmosphere and small changes in soil organic C stocks could have severe impacts on the global C cycle (Knops & Bradley, 2009; Wen-Jie et al., 2011; Cong et al., 2014; Walter et al., 2016; Luo et al., 2017). In addition, documented evidences revealed that the amount of nitrogen present in the soil and in the vegetation is strongly linked to ecosystem fluxes (Field & Mooney, 1986; Reich et al., 2006; Xia, Niu & Wan, 2009; Huff, Potts & Hamerlynck, 2015; Wang et al., 2015; Peng et al., 2017; Song et al., 2017; Tanga et al., 2018). Its evolution in time during the primary succession can be rightfully considered an independent variable influencing vegetation activity and biogenic fluxes.

In scenario 2, we hypothesize the prevalence of specific plant physiological traits over the soil developmental stage. The early stage of vegetation in the primary succession is known for the different biological strategy and functional traits, with higher assimilation rate but lower survival rate (Grime, 1977; Grime, Hodgson & Hunt, 1988; Reich, 2014). Some researchers (Kimmins, 1987) suggest that at the beginning of each seral stage, net productivity will generally increase until perhaps the middle or later part of the stage, then level off, and finally decline somewhat as transition to the next stage occurs.

Given these contrasting drivers, in our studied ecosystem, we would like to know what are the relations between soil evolution, vegetation growth form, local climate and the observed patterns of CO2 exchange and carbon accumulation. We expect that the evolution in time of the soil and of the vegetation traits can act as intertwined variables in addition to climate.

Besides this overall question, the specific objectives of this research, which combined gas exchange with soil and vegetation carbon measurements along an alpine range in the Eastern Italian Alps, were the following:Quantification of the organic carbon and nitrogen accumulated in the soil and in belowground (BG) and aboveground (AG) vegetation along a glacier retreat chronosequence.

Quantification of the net ecosystem exchanges (NEE), of the gross ecosystem exchange (GEE) and of the total ecosystem respiration (Reco) in the different stages of the chronosequence.

Assessment of the linkage between the degree of soil and vegetation evolution and the observed biological fluxes.

Understanding the variables involved in the process of carbon sequestration will help to model the vegetation regrowth after glacier retreat in similar alpine ecosystems, and also to build scenarios where the vegetation green-up after deglaciation is taken into account.

Materials and Methods

Study site

The study sites were in the upper catchment area of the Matsch valley, which is situated within a glacier retreated area in the north of Italy (Bozen/Bolzano in the Italian Alps, Fig. S1). Mean annual air temperature and precipitation (the mean values from the long-term observation 1970–2000), recorded at the closest weather station (Hydrographic Office of the Autonomous Province of Bozen-Bolzano) were 6.6 °C, and 550 mm at 1,570 m a.s.l., where at 2,000 m a.s.l. precipitation increased up to 800–1,000 mm (Penna et al., 2014). Further details about geomorphology and glacial cover are reported in Habler, Thöni & Grasemann (2009), Knoll & Kerschner (2009) and Varolo et al. (2016).

Three study sites were set up in Val Mazia/Matschertal along three transects. The first transect (T1; early successional stage) was established at 2,800 m a.s.l., next to the present-day glacier tongue (glacier 2,800 m). The second transect (T2; intermediate successional stage) was established at 2,450 m a.s.l., but within the glacier extent of the “Little Ice Age” (LIA), therefore covered by the ice until the year 1860 (Moraine LIA 2,450 m). The last transect (late successional stage) was conducted at 2,350 m a.s.l., in an area that was outside the glacier tongue during the LIA which ended in the 19th century (T3, 2,350 m).

After a preliminary botanical survey, used to define the main plant species and communities present on the area, we used photographs taken on the collars to define the plant composition inside each collar. For plant identification, nomenclature and life form classification, we followed Zangheri (1976) and Dalla Fior (1981). For lichen identification and nomenclature, we followed the standard protocol of Smith et al. (2009) and Wirth, Hauck & Schulz (2013), and for moss we followed Frey et al. (2006). Vascular plant species were assigned to taxonomic groups, i.e., monocotyledons and dicotyledons and to plant life forms (Raunkiaer, 1934), i.e., Hemicryptophytes and Chamaephytes.

Gas exchange measurements

The experimental part of the research activity was carried out in three main distinct phases. After a preliminary survey in June, needed to identify transects to be sampled and to place the collars necessary to perform the gas exchange measurements, the first survey was done during the days 9th–11th of July. During these days, the first cycles of gas exchange measurements were conducted. The second cycle of measurements were conducted, with similar methods, in the period August 6–8; the third and last measurement cycle was performed in the period September 17–19.

The gas exchange measurements were done with the use of an automated closed transparent chamber (LI 8100-104C) attached to the analyzer LI 8100 (Li-Cor Biosciences, Lincoln, NE, USA). The measurements (see Varolo et al., 2016; Galvagno et al., 2017; Pavelka et al., 2018; Zhao et al., 2018 for comparison and evaluation of the technique) were conducted on 15 numbered iron collars, 20 cm diameter, divided into groups of five plots (C1; …C5) in each of the three transects. To understand the response of the selected small ecosystem to solar radiation, Nylon filters having the same dimension of the collar were inserted in increasing number inside the chamber, in order to progressively intercept the solar radiation. Measurements were done in sequence, with an increasing number of filters (0, 1, 2, 3, 4, 5, 6, 7, 9, 11 and again zero filters). The final measurement, analog to the first in terms of radiation intensity (full light) was done to show a possible variation in the ecosystem response to light by reason of the temperature variation induced by the presence of filters. Each measurement lasted 60 s, of which 20 s were considered as a dead band and 40 s used to assess the variation in time of the CO2 dry mole density inside the chamber. Solar radiation, as expressed both in terms of energy and photon flux density, was assessed both inside and outside the filters by two distinct radiation sensors (Skye Instruments, Powis, UK). The combined use of sensors placed inside and outside the collar allowed the quantification of the reduction of light intensity caused by the progressive number of shading filters.

Soil and vegetation sampling and sample preparations

Following the final flux measurements, soil biometric sampling was carried out. The sampling was conducted on the entire AG vegetation and the soil below the collar up to 15 cm deep in the soil core. At each transect, five soil samples at three different depths, each one five cm deep (0–5, 5–10 and 10–15 cm) were sampled (n = 15 per transect and each depth interval), and total AG biomass was collected from each collar. A steel ring was used to collect the soil samples. Depending on stone characteristics, the ring was hammered inside the soil, and the stones placed around the ring were broken by using a lump hammer and a chisel or removed by a small shovel. Each sample was labeled and placed in a plastic bag for subsequent desiccation and laboratory analyses. The exact volume of the sample was determined by using plastic balls which were placed inside the collars. The plastic balls’ volume was computed as a function of the balls’ weight, which was measured at each sampling (American Society for Testing and Material (ASTM), 1992).

In the laboratory, to compute soil moisture, all soil samples were weighed and then oven-dried at 105 ± 5 °C until the sample reached a stable weight. Afterward, the bulk density of the sampled soil was evaluated on the basis of measured volume and wet and dry soil density.

Sample analysis

The two portions of soil (fine soil and stones) and the roots were then brought to the laboratory, weighed and the dry fraction of soil and stones was determined. Sub-samples of stones and soil were placed separately in a mortar. Once pulverized, the material was analyzed. Roots were collected, weighed and analyzed separately.

The overall C content in the sampled volume was determined as the sum of the relative concentration of carbon, multiplied by the dry weight (DW) of roots, stones and fine earth. The soil was oven-dried and then weighed and acidified with Hydrochloric acid to eliminate the carbonate present (Brodie et al., 2011). The biomass and soil samples were analyzed for total carbon and nitrogen content and for the carbon isotopic ratio (δ13 C) by a FlashEA™ 1112 Elemental Analyzer (Thermo Fisher Scientific, Waltham, MA, USA).

Statistical analysis

The NEE value (μmol CO2 m−2 s−1) was obtained directly by the transparent chamber during the field measurement campaigns (July, August and September), for each transect. The partitioning of NEE into GEE and Reco was assessed based on the well-established mathematical modeling approach of light response curves. This approach was adapted to be suitable for measurements in small portions of the ecosystem, so as to include soil respiration. Afterward, the values were used to obtain the mean daily dynamic course of NEE, GEE and Reco (μmol m2 s−1). Analyses of residuals demonstrated that linear regression was frequently not appropriate for the determination of CO2 fluxes by closed chamber methods, even if closure times were kept short (Kutzbach et al., 2007). We therefore used the recommended (Li-Cor, 2010) non-linear regression model.

(1) C′(t)=Cx′+(C0′−Cx′)e−α(t−t0)

where C′(t) is the instantaneous water-corrected chamber CO2 mole fraction, C0’ is the value of C′(t) when the chamber closed and Cx’ is a parameter that defines the asymptote, all in µmol CO2 per mol dry air (µmol mol−1); α is a parameter that defines the curvature of the fit (s−1).

To fit carbon assimilation rates with light response curves, i.e., to verify the apparent response to light between PAR and measured NEE with transparent chamber in the three transects (T1, T2 and T3) in the months of July, August and September 2014, we applied a logistic sigmoid model (Moffat, 2012; Eugster et al., 2010).

(2) NEE=2⋅F∞(0.5−11+exp(−2⋅α⋅PPFDF∞))+Rd

where PPFD (photosynthetic photon flux density) is the driving variable, and α (initial quantum yield, alfa), F∞ (maximum NEE at full light) and Rd (daytime ecosystem respiration) are the three model parameters.

Net ecosystem exchange and GEE at the PPFD of 2,000 µmol m−2 s−1 (NEE2000, GEE2000) were calculated based on Eq. (2). A detailed description of this calculation can be found in Schmitt et al. (2010). The Reco at a reference temperature of 10 °C (R10) was assessed using an Arrhenius type model (Lloyd & Taylor, 1994).

(3) Reco=Rref⋅e[E0(0.0178507−1T+46.02]

where Rref is the respiration flux at the constant reference temperature T0 = 10 °C, E0 is an empirical parameter which indicates the temperature sensitivity of Reco, and T is the air temperature in °C.

The flux differences between transects in each measurement campaign were determined by conducting a one-way ANOVA (Tukey b, p < 0.05). Further, to examine the effect of transects on the accumulated C and N in the soil, roots and AG vegetation, a one-way ANOVA (Tukey b, p < 0.05) was used.

The linkage between NEE, GEE and Reco and environmental variables (such as AG biomass (DW), nitrogen content in AG vegetation, carbon content in AG vegetation, BG biomass (DW), nitrogen content in BG vegetation, carbon content in BG-vegetation, soil nitrogen, soil carbon) was firstly tested by multiple linear regression. Preliminary backward deletion procedure, following the Akaike information criterion (AIC), was applied to identify the most relevant variables influencing the biological fluxes. Then, the relaimpo R package was used to build modeling equations.

Then, mixed-effects models (MMs) fitted by restricted maximum likelihood were employed to consider the nesting effects related to the experimental design. MMs were built using “lmer” function in the R package lme4 (Bates et al., 2015; Zuur et al., 2009). Collinearity among all the explanatory variables was determined using variance inflation factors (VIF) to select the proper fixed effects (package MASS, Venables & Ripley, 2002). Sampling site nested in transect were considered as random effect. Model simplification was based on the p-value of the predictors obtained with the function “anova” in R package “car” (Fox & Weisberg, 2019). Aware about the contradictory meaning of p-values in MMs, and the warnings of using them with care (Zuur & Ieno, 2016), final model comparison for each flux was carried out applying the function “anova” in lme4 and using AIC to determine the best model. All statistical computation as well as the figures were produced within the R statistical software environment (R Core Team, 2013, 2019).

Study permission

The Forest Service of the Autonomous Province of Bolzano authorized the researchers to reach the site and to take the samples, permit number V 12 183.

Results

Plant species and vegetation

We have characterized a total of 19 species within collars (Table 1; images of the collars are reported in Fig. S2). According to our identification, the T1 study site, incompletely covered by vegetation, was characterized by one moss and one vascular plant species (Pohlia filum and Poa laxa, none of which was present in the other transects. T2 and T3 sites were fully covered by vegetation. T2 was characterized by 13 species, six of which were in common with T3. Four additional species at that site made a total of 10 species there. In T2, the most common species were Festuca halleri, Lotus alpinus and the moss Racomitrium canescens, while in T3 the most common species were Loiseleuria procumbens and Potentilla erecta. Accordingly, the vegetation could be characterized as an alpine scree community of the Androsacion alpinae alliance (T1), as initial stages of alpine grassland of the Caricion curvulae alliance (T2) and as wind-exposed ridges Loiseleurio-Vaccinion alliance (T3) (Leuschner & Ellenberg, 2017). Full details on species composition in each collar are given in Table 1.

Table 1 Plant species composition in the different transects.

Plant species	Taxonomic group	Growth form	T1	T2	T3	
C1	C2	C3	C4	C5	C1	C2	C3	C4	C5	C1	C2	C3	C4	C5	
Arenaria biflora L.	M	C									+							
Cardamine resedifolia L.	D	H							+					+			+	
Cerastium cerastioides (L.) Britton	D	C									+							
Euphrasia rostkoviana Hayne	D	T						+		+		+				+		
Festuca halleri All.	M	H						+	+	+	+	+	+	+		+		
Flavocetraria cucullata (Bellardi) Kärnefelt & A.Thell	L																+	
Gentiana acaulis L.	D	H							+			+						
Loiseleuria procumbens (L.) Desv.	D	C											+	+	+	+	+	
Lotus alpinus (DC.) Ramond	D	H						+		+	+	+				+		
Luzula alpino-pilosa (Chaix) Breistr.	M	H						+					+	+	+			
Nardus stricta L.	M	H								+								
Poa laxa Haenke	M	H	+	+		+	+											
Pohlia filum (Schimp.) Martensson	B		+	+	+	+	+											
Potentilla erecta (L.) Räusch.	D	H								+			+	+	+	+	+	
Pulsatilla vernalis (L.) Mill.	D	G, H											+		+	+		
Racomitrium canescens (Hedw.) Brid.	B							+	+	+	+	+						
Sibbaldia procumbens L.	D	H							+	+	+							
Soldanella alpinaL.	D	H												+	+		+	
Trifolium alpinum L.	D	H							+									
Total number of plant species in each collar			2	2	1	2	2	5	6	7	6	5	5	6	5	6	5	
Total number of plant species in each transect			2	13	10	
Note:

Different transects (T1:T3) and different collars in each transect (C1:C5). Plant taxa: B, Bryophytes; D, Dycotyledons; L, Lichens; M, Monocotyledons. Life forms: C, Camaephyte; G, Geophyte; H, Hemicryptophyte; T, Therophyte.

Flux parameters patterns

The mean flux parameters (±SD, NEE, GEE and Reco) that were measured during the field campaigns (July, August and September, 2014) in the different stages of the chronosequence (are presented in Fig. 2. Negative flux represents CO2 uptake and a positive one the reverse. In T1, T2 and T3, the mean daily NEE was −1.2 ± 0.9 (mean ± SD), −7.6 ± 0.6 and −6.3 ± 0.4 g C m2 day−1; the mean daily GEE was −2.1 ± 1.3, −11.7 ± 0.6, −10.4 ± 1.1 g C m2 day−1; and the mean daily Reco was 0.8 ± 0.6, 4.1 ± 1.1, 4.1 ± 1.0 g C m2 day−1 across the transects (Fig. 2). The complete dataset is reported in Dataset S1. Figure 2 shows that the daily mean value of NEE, GEE and Reco in the earlier stage of succession (T1) is lower than T2 and T3, significantly.

Figure 2 The mean daily value of NEE, GEE and Reco in the different transects.

Reco, ecosystem respiration; GEE, gross primary productivity; NEE, net ecosystem exchange. Each value is the mean of three campaigns with standard deviation (SD). Statistical significance between transect indicates with lowercase letters for Reco, red capital letters for NEE and blue capital letters for GEE. The different letters are indicating significant differences between transects according to ANOVA test.

During the vegetation seasons in the different stages of the chronosequence, NEE response to PPFD in T2 and T3 showed similar trends and increased with increasing PPFD, whilst, this response was almost negligible (particularly in July) in the earlier stages of succession (Fig. 3). Figure 3 shows that the light response curve line for NEE in T2 was located lower than T3 Additionally, the maximal negative value of NEE in T3 (July = −6.5, August = −6.5 and September = −5.9 µmol m−2 s−1) was lower than T2 (July = −7.4, August = −6.9 and September = −8.0 µmol m−2 s−1), which demonstrates the greater CO2 sink capacity in T2.

Figure 3 Light response curves between PPPFD and NEE in the three transects (T1, 2,800 m; T2, 2,450 m; T3, 2,350 m) in the months of July, August and September, 2014.

(A) July, (B) August and (C) September. NEE, net ecosystem exchange; PPFD, photosynthetic photon flux density.

For comparison across the chronosequence, we mainly followed the indication of Schmitt et al. (2010), and we used NEE2000, GEE2000 and the Reco at a reference temperature of 10 °C (R10). The mean NEE, GEE and alfa as measured during the field campaigns were the highest for the LIA moraine, significantly, and the lowest for the glacier (Fig. 4). Across the chronosequence, the highest mean value of Reco was recorded for T2 during July. Conversely during August and September, it was highest in T3 (Fig. 4). Reco reached the highest mean value in August (T1 = 1.6 ± 0.9, T2 = 4.9 ± 1.7 and T3 = 5.5 ± 1.4 µmol m−2 s−1), which was the warmer period during the study (T1 = 13.6 ± 1.3, T2 = 13.0 ± 1.7 and T3 = 16.8 ± 2.4 °C), whereas the lowest value was observed in September, a period with the lowest air temperature during the study (T1 = 6.2 ± 1.5, T2 = 9.9 ± 1.8 and T3 = 10.4 ± 1.2 °C).

Figure 4 Mean flux parameters at the three transects during the field campaigns.

(A) Mean NEE2000. (B) GEE2000. (C) Reco at a reference temperature of 10 °C (R10). (D) alfa. NEE2000—net ecosystem exchange at the PPFD of 2,000 μmol m−2 s−1; GEE2000—gross primary productivity at the photosynthetic active photon flux density (PPFD) of 2,000 μmol m−2 s−1; Reco, ecosystem respiration; α, the quantum yield in the light response curve. Negative values indicate net CO2 uptake and positive values represent net released CO2 to the atmosphere. T1 = 2,800 m, T2 = 2,450 m, T3 = 2,350 m. Asterisks indicate significance levels: *p ≤ 0.05, **p ≤ 0.01, ***p ≤ 0.001.

For all the three transects, the highest mean value of NEE was observed in September (T1 = −5.9 ± 4.3, T2 = −11.3 ± 6.6 and T3 = −8.0 ± 8.3 µmol m−2 s−1 and the most negative value recorded in August for T2 and T3 (T2= −8.5 ± 4.4, T3 = −6.4 ± 0.7 µmol m−2 s−1) and July for T1 (−1.9 ± 3.8 µmol m−2 s−1). September and July showed a much higher value of NEE for T2 and T3 (Fig. 4). The highest mean GEE value was −6.6 ± 4.2, −15.6 ± 5.0 and −12 ± 3.0 µmol m−2 s−1 in T1, T2, T3, respectively. The greatest amplitude of GEE occurred in July for T2 and T3, and in September for T1.

Organic carbon and nitrogen content

We calculated the mean (±SD) accumulated C and N content in the soil, roots and AG biomass in the three transects (Fig. 5; Table 2; full results of the analyses are reported in Dataset S2). The amount of C in the soil was 40 ± 12 g m−2, 557 ± 115 g m−2 and 58 ± 10 g and 384 ± 98 g m−2 in T1, T2 and T3, respectively, while N was 22 ± 3.4, 58 ± 10 and 59 ± 8.9 g m−2 in the same transects. AG biomass, C and N mass increased across the chronosequence but was not significantly different between transects (Table 2). Moreover, C and N mass and AG biomass in T2 were higher than T1 and T3 (p > 0.05, Table 2).

Figure 5 Boxplots showing the results of soil and vegetation analyses.

(A) Soil carbon. (B) Soil nitrogen. (C) Carbon in BG vegetation. (D) Carbon in AG vegetation. (E) Nitrogen in BG vegetation. (F) Nitrogen in AG vegetation. (G) BG biomass. (H) AG biomass. Lowercase letters indicate the results of statistical tests with different letters indicating significant differences between transects. BG, belowground; AG, aboveground.

Table 2 Accumulated carbon and nitrogen in soil, roots and AG vegetation (g m−2) in each transect.

Parameter (g m−2)	T1	T2	T3	p-Value	
AG vegetation	Biomass	207 ± 161	1,753 ± 2,101	1,095 ± 170	0.175	
C mass	66 ± 49	752 ± 866	554 ± 100	0.128	
N mass	1.5 ± 1.0	24 ± 30	11 ± 2.0	0.164	
BG vegetation	Biomass	58 ± 78	1,774 ± 385	2,062 ± 411	***	
C mass	24 ± 34	753 ± 157	959 ± 183	***	
N mass	0.36 ± 0.5	20.0 ± 5.4	20.4 ± 6.3	***	
Soil	C mass	40 ± 12	557 ± 115	384 ± 98	***	
N mass	22 ± 3.4	58 ± 10	59 ± 8.9	***	
Notes:

T1, 2,800 m; T2, 2,450 m; T3, 2,350 m; ±SD, standard deviation; AG, aboveground; BG, belowground.

*** p ≤ 0.001 indicate significance levels.

The accumulated biomass, C and N mass in the BG vegetation (roots and rhizomes) increased significantly during the ecosystem development. Observed biomass, C and N mass in the BG vegetation was 58 ± 78, 1,774 ± 385 and 2,062 ± 411g m−2; and 24 ± 34, 753 ± 157 and 959 ± 183 g C m−2 0.36 ± 0.5, 20.0 ± 5.4 and 2.4 ± 6.3 g N m−2 in the three transects (Table 2). The biomass and C mass of BG vegetation were higher at T3 than at T1 and T2, significantly (Fig. 5; Table 1), whilst N in BG vegetation was significantly higher at LIA moraine than glacier and conoid (Fig. 5; Table 2).

The majority of C, N and biomass were accumulated in BG vegetation while C, N and biomass accumulation AG were lower (Figs. 3D–3H and 5C). In T1 and T2, no significant differences can be seen between AG and BG C, N and biomass but in T3 accumulated C, N and biomass in roots were significantly higher than AG vegetation (biomass p = 0.001, carbon p = 0.002, nitrogen p = 0.01).

Soil profile analyses

Carbon and nitrogen content increased across the chronosequence in the soil profile significantly (p < 0.001). The highest values of C and N content were found in the conoid 2,350 m (C = 7,770 and N = 443 g m−2) and the lowest value was found in the glacier 2,800 m (C = 861 and N = 96 g m−2, Fig. 6). The majority of C and N accumulation were observed in the first level of soil profile near the soil surface, in the top five cm of the soil profile (T1 = 70.0%, T2 = 44.6%, T3 = 61.3% for C; and T1 = 54.6%, T2 = 44.5% and T3 = 55.0% for N). Soil C and N mass in this depth (~ −2.5 cm) of the soil profile was significantly greater than at other soil depths (~ −7.5, ~ −12.5, p < 0.001). With increasing soil depth, C and N mass in the soil profile decreased significantly (Fig. 6, p < 0.001).

Figure 6 Average vertical profiles of carbon, nitrogen and bulk density in the three transects.

(A) Carbon. (B) Nitrogen. (C) Soil bulk density. Average values in the different heights (±2.5 cm) of vegetation and soil profiles.

At ~2.5 cm above the soil profile, the total C and N contents were significantly different (p < 0.001). The highest amount of C and N were found in the LIA moraine 2,550 m (C = 1,177 g C m−2 and N = 38 g N m−2) and the lowest amount was found in the glacier 2,800 m (C = 114 g C m−2 and N = 2.63 g N m−2, Fig. 6).

Soil bulk density decreased along the glacier retreat chronosequence. The highest value of bulk density was found in the glacier 2,800 m (1.76 kg dm−3) and the lowest value was found in the LIA moraine 2,550 m (1.41 kg dm−3). Soil bulk density increased with increasing soil depth significantly (T1 p = 0.02, T2 p < 0.001, T3 p < 0.001). The lowest value was found near the soil surface at the ~ −2.5 cm depth of soil profile (T1 = 1.42, T2 = 0.66 and T3 = 0.63 kg dm−3), while the highest value was at a depth of about −12.5 cm (T1 = 1.99, T2 = 1.84 and T3 = 1.94 kg dm−3, Fig. 6C). In the top level of the soil profile (~ −2.5 cm), the bulk density was significantly changed across the chronosequence (p = 0.007) but this change was not significant in level 2 (~ −7.5 cm, p = 0.27) and level 3 of the soil profile (~12.5 cm, p = 0.44).

Finally, we provide an assessment of the net production (µmol m−2 s−1) and mean accumulated C (total AG, BG and soil C, kg m−2) along the chronosequence of the primary succession after glacier retreat. As shown in Fig. 7, the highest net production and total C was observed in T2 and the lowest in T1. This worthy and peculiar finding is indicating that the net production and total C (AG, BG and soil) approached their maximum value in the middle ecosystem (T2) instead of the older ecosystem (T3).

Figure 7 Schematic illustration of the net production and total C along the glacier retreat chronosequence.

Net production (µmol m−2 s−1) and total C (mean accumulated carbon in AG and BG vegetation + soil carbon (kg m−2), along the glacier retreat chronosequence. GPP, gross primary production = GEE; Reco, ecosystem respiration; T1, 2,800 m; T2, 2,450 m; T3, 2,350 m).

Assessment of the linkage between soil and vegetation evolution and the observed biological fluxes

The linkage between NEE, GEE and Reco and site variables (AG and BG biomass, C and N, soil C and N, air temperature) was tested by multiple linear regression and by mixed models (Dataset S3) In the multiple linear regression assessment, we found that, over the whole season, the flux better explained by site conditions was Reco (80.57% variance explained by five regressors: C AG, DW BG, C BG, Nsoil and Tair,), while GEE and NEE variability was explained to a lower extent (54.82% and 41.19%, respectively) by a reduced set of variables (DW BG and C soil in both cases, and Tair only for NEE, see Table 3).

Table 3 Multiple linear regression models developed to explain the observed fluxes using all environmental variables.

Parameter	Developed model	Multiple R-squared	p-Value	SE	
NEE	NEE = −6.125023*−0.002048 DW BG −0.008529 Csoil + 0.421186 Tair	0.4119	0.0001076	4.197	
GEE	GEE = −2.975012*−0.001984*DW BG −0.013756**Csoil	0.5482	<0.0001	4.361	
Reco	Reco = −2.9809733***−0.000953*C AG + 0.0061722**DW BG −0.0132679**C BG + 0.0722129*** Nsoil + 0.2600114***Tair	0.8057	<0.0001	0.9601	
Notes:

DW BG, dry weight of BG vegetation; Csoil, soil carbon; C BG, carbon content in BG vegetation; Nsoil, soil nitrogen; Tair, air temperature; SE, residual standard error. The effect of nested sampling points is not considered.

Significance levels: value without asterisk 0.1 ≤ p ≤ 0.05.

* p ≤ 0.05.

** p ≤ 0.01.

*** p ≤ 0.001.

In the preliminary analysis done for the mixed model assessment, we found that many of the explanatory variables were correlated, such as DW AG and C BG. After removing the variables with the highest VIF scores, we determined that the best variables for use in the mixed effect models were DW BG, DW AG, Tair and C_soil. NEE was significantly negatively affected by DW BG (χ2 = 4.11, p = 0.043). GEE was significantly negatively affected by C soil (χ2 = 8.94, p = 0.003). Reco was significantly positively affected by C soil (χ2 = 37.83, p < 0.001) and Tair (χ2 = 32.70, p < 0.001). Refer to Table 4 for intercept estimates.

Table 4 Best mixed-effect models developed to explain the variation in NEE, GEE and Reco.

Parameter	Developed model	SE	χ squared	χ squared p-value	AIC	
NEE	NEE = −5.263963−0.002 DW BG −0.007591 C soil + 0.396281 Tair	Intercept = 2.677909
DW BG = −0.0011350
Csoil = −0.005270
Tair = 0.396281	4.11
2.08
3.58	0.04
0.15
0.06	255.2	
GEE	GEE = −2.956291−0.002016 DW BG −0.013688 Csoil	Intercept = 1.279064;
DW BG = −0.01134;
Csoil = −0.004578	3.16
8.94	0.08
0.003	257.6	
Reco	Reco = −1.674835 + 0.005646 C soil 005646 + 0.266757 Tair	Intercept = 0.586761;
C soil = 0.000918;
Tair = 0.046647	37.83
32.70	0.001
0.001	134.6	
Note:

Models were developed starting from the reduced set of explanatory variables after VIF selection and further removing those with higher p-value until the lower AIC was obtained. The effect of nested sampling is taken into account considering the number of plots nested into the transects as random effect. Significance of models tested against the null model using Type 2 Wald χ square tests.

Discussion

Flux parameter pattern

Our results indicate that the T2 and T3, as intermediate and older successional stages, acted as a larger C sink, with NEE, GEE, α and Reco significantly higher than T1 (earlier successional stage). This finding is in line with those of Nakatsubo et al. (1998, 2005), Bekku et al. (2004), Guelland et al. (2012) and D’Amico et al. (2014), who found the higher flux parameters in the later stages of succession.

All the flux parameters related to assimilation, namely NEE, GEE and α, approached their highest value in T2 during all three field campaigns, although for some values the difference with T3 was not significant (Fig. 2). Meanwhile, the highest mean value for Reco was recorded in T3 during August and September (Fig. 4, p > 0.05). The higher flux parameters, and particularly the quantum yield of assimilation (α) in T2, are consistent with the well-established positive relationship between species richness and productivity (Waide et al., 1999).

The observed higher Reco in T3 can be explained by the higher BG biomass, C and N content (Table 2; Fig. 5), which may result in a higher autotrophic respiration (Bekku et al., 2004; Guelland et al., 2012). On the other hand, it is well known that temperature has a strong effect on soil and plant respiration, and generally, the soil respiration rate would increase under future global warming (Oechel & Vourlitis, 1994; Bekku et al., 2004; Eugster et al., 2010). In all three transects, Reco reached the highest mean value in August and lowest in September, respectively, the warmest and coldest periods during the study. Apparently, the greater Reco led to a lower NEE. In fact, NEE reached the highest value in the coldest period (September) and the lowest value in the warmest period (August).

Organic carbon and nitrogen content

We quantified the organic C and N accumulated in soil, BG and AG vegetation in the different stages of the primary succession, as one of our objectives (Fig. 5; Table 2). We found an increase of organic C and N accumulation in soil, BG and AG vegetation with ecosystem development, as has been recorded by several researchers in recently deglaciated terrain in the Alps (Matthews, 1992; Egli, Fitze & Mirabella, 2001; Burga et al., 2009; Mavris et al., 2010; Dümig, Smittenberg & Kögel-Knabner, 2011; Kabala & Zapart, 2012; D’Amico et al., 2014; Li et al., 2018), China (He & Tang, 2008) and high arctic regions (Nakatsubo et al., 1998, 2005; Bekku et al., 2004; Yoshitake et al., 2011; Osono et al., 2016). However, the increment in ecosystem carbon content was evident only in recently deglaciated terrain (T2), with an average increment of 9 g C m−2 year−1 in the ~150 years after deglaciation, while further C accumulation was negligible in the older successional stage (T3).

In a broader temporal perspective, we expect that, forest vegetation will be able to establish and grow. For the forest establishment, however, several conditions must become effective, like the foreseen increase in temperature, the soil development and the dispersal of tree seeds. In addition, the sheep-farming activity should be reduced. If all these conditions will be met, the current limited capacity of accumulating carbon will be only transient, since forest vegetation is known for having generally larger and long-lasting capacity of accumulating carbon (Luyssaert et al., 2008).

While the differences in organic C accumulated AG were not significant, biomass in BG vegetation was significantly greater in older stages (T2 and T3) than in the early stage (Fig. 5; Table 2). These results suggest that the growth rate of BG parts in T3 (p < 0.05) and AG parts in T2 (p > 0.05) are considerable.

The deeper soil in older stages can partly account for the greater BG biomass compared to early stages, where the poor development of soil, extremely low air temperature, low soil moisture and short growing season might limit the accumulation of BG biomass (Chapin et al., 1992; Oechel & Billings, 1992; Osono et al., 2016). Therefore, BG parts of plants are almost negligible in the earlier stages of succession (Nakatsubo et al., 1998). It is probable that the highest AG biomass, C and N in T2 is due to the greatest photosynthetic and carbon sink capacity that was observed in this stage.

The soil C and N content tended to increase along the chronosequence. Soil N reached the highest value in T3, but the difference with T2 was insignificant. The observed range of soil C and N was 40–557 g C m−2 and 22–59 g N m−2, which were within the ranges reported by other studies in the deglaciated terrain (Egli, Fitze & Mirabella, 2001; Dümig, Smittenberg & Kögel-Knabner, 2011; Kabala & Zapart, 2012), and lower compared to amounts reported by He & Tang (2008), Burga et al. (2009); Mavris et al. (2010) and Yoshitake et al. (2011).

As expected, the highest values of C and N content were found in the top five cm of the soil profile, where the largest amount of organic matter from litter accumulates (Egli, Fitze & Mirabella, 2001; Pei-qin et al., 2005; Kabala & Zapart, 2012; Vogel, 2013). In contrast to C and N, bulk density decreased with time since deglaciation and increased with soil depth as found in previous studies (Schrumpf et al., 2011). Similarly, other studies have reported a decrease in bulk density with time since deglaciation and an increase with soil depth, with accumulation of organic matter, the start of parent material alteration, increase in ion exchange capacity and the formation of weathering products (Egli et al., 2006; Bernasconi et al., 2011; Vilmundardóttir, Gísladóttir & Lal, 2015).

The linkage between soil and vegetation evolution and the observed biological fluxes

Plant C and soil organic carbon are in active exchange with the atmosphere and small changes in soil organic C stocks could have severe impacts on the global C cycle (Knops & Bradley, 2009; Wen-Jie et al., 2011; Cong et al., 2014; Walter et al., 2016; Luo et al., 2017). In general, only a few features have been considered, such as the microbiological contribution to the carbon cycle (Egli et al., 2006; Bardgett et al., 2007; He & Tang, 2008; Mavris et al., 2010; Dümig, Smittenberg & Kögel-Knabner, 2011; Guelland et al., 2012; Kabala & Zapart, 2012; Chae et al., 2016). Most of these studies found a clear increase in the flux parameters with increasing ecosystem age. In particular, flux parameters were suggested to be linked to variables increasing along the deglaciation chronosequence, like vegetation cover, plant productivity, biomass, C and N. This pattern was observed both in the Alps (Matthews, 1992; Egli, Fitze & Mirabella, 2001; Burga et al., 2009; D’Amico et al., 2014; Li et al., 2018) and in high arctic regions (Nakatsubo et al., 1998, 2005; Bekku et al., 2004; Yoshitake et al., 2011). Further, these studies pointed out that the organic matter increased mainly within the uppermost part of the soil profile, and decreased with increasing depth, whilst, bulk density decreased with time since deglaciation (Egli, Fitze & Mirabella, 2001; Egli et al., 2006; Pei-qin et al., 2005; Kabala & Zapart, 2012; Vilmundardóttir, Gísladóttir & Lal, 2015). However, Wietrzyk et al. (2018) highlighted that vegetation cover is the main factor which affects soil properties during primary succession. Additionally, chemical soil properties and distance from the glacier foreground affect species distribution and vegetation cover.

In our study, we confirmed the increase in fluxes (GEE, Reco) and soil C and N only during the first period after deglaciation, while in the later stage most of the measured variables leveled off or even decreased. We should consider that the organic matter, a few years after its formation, tends to be stabilized in the soil profiles, and its role in soil fertility is strongly reduced. Recent global radiocarbon studies suggest a higher average age for the soil carbon than previously thought (He et al., 2016). We can also assume that interaction between organic matter degradation and soil fertility are positively related, although the topic is still a matter of debate (Janzen, 2006; Bradford et al., 2016).

To test the linkage between soil and climate variables with observed fluxes, we firstly applied a multiple linear regression. Then, we applied MMs (Harrison et al., 2018) to properly evaluate the results arising from a nested scheme of the sampling plots. The outputs show similarities, with both modeling approaches selecting the same explanatory variables within the NEE and GEE models. With regards the mixed model for Reco, the variables were different with only Tair remaining in the selected variables, along with C soil. In our developed mixed model, Reco was positively affected by both Tair and Csoil. It is well established that Reco increases with temperature with an exponential (Q10) or an Arrhenius type function (Lloyd & Taylor, 1994; Uchida et al., 2002; Janssens et al., 2003) and also with C soil (Migliavacca et al., 2011). We must highlight that the nested scheme of the sampling plots used in our study, based on the long-term tradition of transects, is not ideal from the modern statistics point of view, although easier to implement for practical reasons. We recommend that, whenever possible, a complete random selection of the sampling sites is performed, in accordance with the most recent vegetation and soil sampling schemes (Saunders et al., 2018).

Other variables were loosely related with fluxes, with below-significance correlation: In particular, we did not find significant correlation between AG N and biological fluxes. This result can be partly explained by the non-linear response of biological fluxes to N. For instance, Peng et al. (2017) showed that gross ecosystem productivity, Reco and NEE all exhibited nonlinear responses to increasing N additions. Furthermore, we should emphasize that much of the stored N in the observed vegetation type is presumably not directly used for the current year photosynthesis, but it represents conversely a long-term investment to enhance plant resilience to the frequent disturbances.

We found also that the different Raunkiaer classification of the different life forms is not adequate for understanding the different plant growth strategies and forecast the fluxes. In fact, we observed a convergence in plant allocation of organic biomass, irrespective to the Raunkiaer classification (Table 3). For instance, the dominant plant in T3, i.e., Loiseleuria procumbens, is a plant with large lignified underground tissue, although it is classified among Camaephytes based on its buds position. Interestingly, Caccianiga et al. (2006) found a switch in plant strategy types (Grime, Hodgson & Hunt, 1988) during primary succession leading from ruderal (R) to stress (S) strategy dominance. We hypothesize that the modest linkage between AG and BG N and observed biological fluxes reflects the dominance of ruderal behavior in mid successional stages, while late successional stages with stress-tolerating strategists could have more investment in BG structures reflected in nitrogen accumulation in plant reservoirs above and below ground in addition to the foliar N content. The linkage between plant properties, in particular AG N, is therefore weaker than what was found by López-Blanco et al. (2018) in tundra ecosystems by analyzing the leaf N content only.

Net production and accumulated biomass along an ecological succession gradient

Kimmins (1987) suggested that at the beginning of each seral stages, net productivity will generally increase until perhaps the middle or later part of the stage, then level off and finally decline somewhat as a transition to the next stage occurs. This pattern was confirmed by our results. T3 has started to develop earlier and thus can be considered as an older succession stage than T2, but despite this fact, our result revealed that net production and mean biomass accumulation peaked in the T2 (Fig. 7). This finding implies that carbon accumulation in the different stages of the primary succession on the Matsch glacier foreground (Alps, Italy) was not ruled only by time since deglaciation. Kimmins (1987) suggested that net productivity declines in the older ecosystem because later successional species have been selected during their evolution more for resilience to adverse conditions than for rapid growth.

These findings have to be placed in the broader context of recurrent disturbances common to periglacial areas, possibly affecting the spatial distribution and the timing of evolution of vegetation. This subject has been discussed by Burga et al. (2009), and they concluded that large-scale factors such as time since deglaciation, topography and disturbance (floods, rockfalls, avalanches), as well as small-scale factors such as grain size and water content of the substrate, micro-relief and micro-climate seem to be crucial for the development of vegetation and soil after deglaciation.

Conclusions

We found an average carbon accumulation along the glacier retreat chronosequence of 9 g m−2 year−1 in the first ≈150 years after deglaciation. The largest part of the carbon was accumulated in BG living organs. After that period, we did not observe significant further C accumulation.

Net ecosystem exchange increased significantly moving from the site near the current glacier tongue (2,800 m a.s.l.) to the site in the LIA moraine (2,450 m a.s.l.). In the oldest site (T3, 2,350 m a.s.l.), the maximal uptake was similar to the one of the LIA moraine, but the quantum yield was lower and Reco was higher.

We observed also a positive linkage between small-scale plant species richness and assimilation at low radiation intensity. Additionally, higher N in the biomass of the Hemicryptophyte life form and soil C were linked with higher Reco. The species richness was maximal at the intermediate stage of ecosystem development after glacier retreat. At this seral stage (≈150 years), the sink capacity exceeded that of the older ecosystem, not influenced by the glacier presence in the last centuries.

This study suggests a strong negative feedback mechanism in the response of global change, where recently established “ruderal” plants sensu Grime (1977; Caccianiga et al., 2006) rapidly expanding after glacier retreat, express a strong carbon sink capacity, therefore effectively counteracting the increasing levels of carbon dioxide concentration in the atmosphere. This high sink capacity is only transitory, since the older seral stage, where more stress-tolerating plants prevail, approaches the steady state condition.

Supplemental Information

Supplemental Information 1 Map of the study area.

T1, T2 and T3 indicate the experimental transects. The continuous black contour line indicates glacier extent in 2011.

Click here for additional data file.

Supplemental Information 2 Images of the collars and of the sampled ecosystem portions.

The letter T indicates the transect (T1–T3) and the letter C indicates the collar (C1–C5).

Click here for additional data file.

Supplemental Information 3 Light response curves original data.

In this file are reported the environmental variables data and those obtained by the CO2 analyzer Li-8100.

Click here for additional data file.

Supplemental Information 4 Results of the soil and vegetation analyses.

In the dataset are reported the raw results of the analysis performed in the different samples.

Click here for additional data file.

Supplemental Information 5 Dataset used for statistical analysis of interactions between soil and plant chemistry and observed fluxes.

Click here for additional data file.

Additional Information and Declarations

Competing Interests

Author Contributions

Field Study Permissions

Data Availability

Leonardo Montagnani is an Academic Editor at PeerJ.

Leonardo Montagnani conceived and designed the experiments, performed the experiments, analyzed the data, authored or reviewed drafts of the paper, approved the final draft.

Aysan Badraghi analyzed the data, prepared figures and/or tables, authored or reviewed drafts of the paper, approved the final draft.

Andrew Francis Speak analyzed the data, prepared figures and/or tables, approved the final draft, contributed to the language editing of the text.

Camilla Wellstein authored or reviewed drafts of the paper, approved the final draft, contributed to the botanical part of the study.

Luigimaria Borruso analyzed the data, approved the final draft.

Stefan Zerbe conceived and designed the experiments, approved the final draft, contributed to the botanical part of the study.

Damiano Zanotelli performed the experiments, analyzed the data, prepared figures and/or tables, approved the final draft, contributed to the statistical part of the study.

The following information was supplied relating to field study approvals (i.e., approving body and any reference numbers):

Field survey was authorized by the Forest Service of the Autonomous Province of Bolzano (Permit number V 12 183).

The following information was supplied regarding data availability:

The raw measurements are available in the Supplemental Files.

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
