# Peer review of "Evidence for a non-linear carbon accumulation pattern along an Alpine glacier retreat chronosequence in Northern Italy"

_PeerJ, doi:10.7717/peerj.7703_

## Round 0.1 · original submission · Minor Revisions

We have received the reports from our advisors on your manuscript.

Based on the advice received, I feel that your manuscript could be considered for publication should you be prepared to incorporate a number of minor revisions. When preparing your revised manuscript, you are asked to carefully consider the reviewer comments which can be found below*, and submit a list of responses to the comments.

·

Basic reporting

The article is clearly written throughout. I have a few comments/suggestions for this category:

• Figure 3 has JPEG compression artefacts and should be re-produced at higher resolution. In addition, the unified scale of Figure 3 visually flattens the response curve in T1 in July… although I’m not sure this is worth re-drafting the figure over.
• Figure 6 has positive soil depth values! I assume these are aboveground vegetation measurements or have missed a key point, but clarification would be welcome.
• Figure 7 legend should probably have a wider legend key to show the distinct line-types.

Experimental design

The authors clearly state their research goals and explain their methodology. I have no major comments for this section, but I would suggest that, in the case of flux measurements, more measurement events during the year would have been welcome and, ideally, would have been performed for several years.

Validity of the findings

The authors clearly present their results and include all underlying data in the supplement. My main concern above about the frequency/duration of measurements does not invalidate their findings in any way, but simply reflects a desire for a more complete picture of the ecosystem.

Additional comments

Some specific comments below. The last comment (regarding lines 502-504) I believe would greatly improve the perceived impact and importance of post-glacial colonization ecosystems to other fields.

• Lines 92 to 97 describe Figure 1, which are three hypothetical flux trajectories following colonization. All three show a similar, fundamental trend of increasing respiration with increasing assimilation of C. However, keeping the fundamental trend in mind (as A increases, R should also increase due to plant maintenance R as well as soil R), more than these three cases can be imagined. In addition, ‘time’ here is clearly meant to be at least somewhat hypothetical, but the x-axis could be labeled with phase of succession rather than arbitrary numbers. Likewise, I would recommend removing the arbitrary numbers from the y-axis altogether.
• Unpacking the above strictly for Scenario 1: The authors claim this scenario depicts a soil-dominated outcome where maximum sink capacity (A-R) is achieved in the later stages, where soil is most fully developed. However, given that soil R is a function of (among other things) the size of the soil C stock, we might imagine that in later stages of succession where a steady-state is achieved with maximum plant cover and fully developed soil, that the sink capacity would actually be less because it is under these conditions where soil R is maximized (due to the large stock).
• I am not looking for a specific response from the authors, as they do mention (lines 121-122) that the overall trajectory is likely to be driven by an interplay between plant and soil development. I would, perhaps, consider emphasizing that the scenarios presented are idealized ‘end-member’ hypotheticals somewhere in the paragraph on lines 92-97.
• Line 151 (and subsequent mentions): The third transect site is named ‘Conoid’, but I don’t quite understand the significant of this moniker? Is this a discipline-specific term? If it is arbitrary, I might suggest using a more descriptive term for the third transect (e.g., ‘old growth’).
• Line 297: This should reference Figure 5, right?
• Line 306-307: This series of estimates of BG stocks is difficult to read. Consider editing or removing and focusing on the reference to the Table.
• Line 374-376: How much of this could be attributed to changes in phenology? I am not familiar with the site type and species, but shifts in plant resource allocation that occur during a season could account for some of this change, right?
• Line 433: This thought has occurred several times during reading… but how certain are you that T3 represents primary succession from T2? Presumably there are not many disturbances that may have occurred, but since there is no age estimate for T3, it may not present the final stage of primary succession for the T2 (or T1, for that matter).
• Line 477: Another thought… if we wanted to calculate total C storage on previously barren land, then we should measure soil C to bedrock, yes? While 0-15 cm may be sufficient for estimating total C storage in T2, this may not be the case for T3, right?
• Line 502-504: I am glad the authors returned to this point. The yearly accumulation flux is fine, but better would be the following: compare the glacial albedo radiative forcing to the radiative forcing of the CO2 captured and stored as C stock within the new ecosystem. This would allow a direct estimation of just how much of the glacial radiative forcing that is lost by retreat is offset by plant colonization. If data are available, a further comparison to forest ecosystem C stocks in the region could provide a forward-thinking estimate if the tree line is actually likely to move into the current study site.

Reviewer 2 ·

Basic reporting

The manuscript is clearly written in good English, with one or two places noted for improvement in the author comments below. Overall the authors include sufficient background and I point out a few places where clarification or more context could be used. The article structure is clear and logical, and the hypotheses for the study are well-defined and clearly related to the results.
In the discussion and introduction it might be interesting to cite some of the ideas regarding soil C development and storage potential brought by (Janzen 2006, Bradford et al. 2016, He et al. 2016).

References:
Bradford, M. A., W. R. Wieder, G. B. Bonan, N. Fierer, P. A. Raymond, and T. W. Crowther. 2016. Managing uncertainty in soil carbon feedbacks to climate change. Nature Climate Change 6:751–758.
He, Y., S. E. Trumbore, M. S. Torn, J. W. Harden, L. J. S. Vaughn, S. D. Allison, and J. T. Randerson. 2016. Radiocarbon constraints imply reduced carbon uptake by soils during the 21st century. Science 353:1419:1424.
Janzen, H. H. 2006. The soil carbon dilemma: Shall we hoard it or use it? Soil Biology and Biochemistry 38:419–424.

Experimental design

The experimental design is adequate with clearly defined and interesting research questions. The investigation is performed to a high standard and the methods and results are clearly presented. In the author comments I point out some ideas for minor improvements.

Validity of the findings

The data are well presented and the findings are placed in context with a relevant discussion of the meaning and implications. I give a few suggestions and more detailed comments in the author comments, below.

I am curious whether there are specific projections for glacial retreat in the Matschertal? If yes, this could be mentioned to provide context for the expected rate of change.

The data are provided with the exception of September fluxes, which seem to be missing from the excel document. In figure S1 I think it would be helpful to show the current glacier extent in relation to the transects. For the data reported in tabular form explicit metadata would be helpful. This can be short and just a description of columns and abbreviations.

Additional comments

Line 53-54: could be more clearly/directly phrased, eg: Glacial expansion in the Alps peaked around 1860 and since then have been retreating leaving an abundance of bare ground…

Line 72-74: does this study measure changes in energy balance? This sentence could be deleted. It is an interesting fact that is not totally relevant to the C cycle focus of the study.

Line 75: use a direct object. What does ‘this phenomenon’ refer to? The phenomenon of glacial retreat or of the radiative forcing discussed in the previous sentence?

Line 81: why the mention of black carbon here?

Line 92-97: this is nice.

Line 102: I think this would read more smoothly if changed to: ‘the variation in carbon and nitrogen is considered to influence photosynthesis and respiration equally’.

Line 103-111: I’m not sure I totally understand the mechanisms for prevailing soil processes? Is that because microbial communities and soil C&N cycling might have a stronger influence on C dynamics than the vegetation? This paragraph talks a lot about plants, it might be better to focus on the soil?

Line 112-118: scenario 3 is laid out very nicely, focused on reasons why plant processes could be the most dominant mechanism for C accumulation.

Line 139: That’s a nice map. Would it be possible to indicate the current glacial extent as well? Maybe a dotted line? Or perhaps a google earth image with T1-T3 indicated for reference?

Line 212: the data provided in the supplement is missing September? I found only July and August. The data is well organized and generally easy to understand. Still, I would having a metadata page that describes the columns and some values. For instance, what are 2b,3b,etc and ‘ck’? One of those is presumably a repeat with no filter. I understand that f0-f11 are increasing filter thickness (decreasing light).
Why was the zero-filter measurement repeated again at the end? I think typically we assume that full photosynthesis might take some time to recover which is why we measure from full to low-light conditions.

Line 192-194: oh, that’s a cool method!

Line 214: modified how? I am not sure what modification would be needed to the light-response method?

Line 227: September data are missing in supplement?

Line 244: This is a nested design, with 5 chambers nested within each transect. Was a nested analysis (mixed effects model) tested? The sample size is quite small so it might be difficult but a mixed model might have greater statistical power because it could account for presumably greater similarity between collars from a single transect.

Line 266-268: are these three numbers presumably per transect? Reported as: T1, T2, T3? Please clarify.

Line 269: earlier stage of succession is T1? Add to text.

Line 280: alfa – should this be the Greek letter alpha? What does alpha signify?

Line 282-283: what are conoid and LIA moraine? Transects?

Line 285: this is a nice way to report transect T1, T2, T3 numbers.

Line 286: was the vegetation already senesced in September?

Line 299-303: conoid and LIA moraine again?

Lime 350-354: were the regressions done on fluxes normalized by light and temperature?

Line 357: could the terms LIA and conoid be explained? Many readers might not know these terms since this is not a journal for glaciologists. And could the terms be linked explicitly to T1, T2, T3?

Line 390: The comment on forest establishment needs a little more context. Is temperature the primary limitation? Do dispersal and soil development play a role in limiting trees?

Line 397: presumably the relationship between BG biomass and soil present a feed-back. As more soil develops more roots can grow, which leads to more soil formation?

Line 423-428: this sentence is a little hard to follow. It might be better to break it up into two, and fix some grammatical errors.

---

## Round 0.2 · Minor Revisions

Line 72-74
I suggest deleting the sentence “This improvement …. (Forzieri et al. 2017)”. In my opinion even if interesting this is only a speculation - very difficult to be assessed.


Line 81
I agree with the referee that mentioning BC here is not completely correct nor useful. BC is an operational definition based on the optical properties of the airborne aerosol and has little to do with the chemistry of the C cycle. I suggest replacing “black carbon and nitrogen” with “carbon and nitrogen compounds”.

---

## Round 0.3 · accepted · Accept

In my opinion the paper can be accepted for publication in its present form.